# Non-Linear Response of Alpha and Beta Diversity of Taxonomic and Functional Groups of Phytoplankton to Environmental Factors in Subtropical Reservoirs

**DOI:** 10.3390/microorganisms12081547

**Published:** 2024-07-29

**Authors:** Zhenlong Xiang, Haiyu Niu, Quehui Tang, Ren Hu

**Affiliations:** Institute of Hydrobiology, Jinan University, Guangzhou 510632, China; xiangzhenlong@163.com (Z.X.); niuniu-nhy@163.com (H.N.); tangquehui@scsfri.ac.cn (Q.T.)

**Keywords:** functional group, alpha–beta diversity, GDM model, environmental variables, geographic distance

## Abstract

Exploring the response of the diversity of phytoplankton species and functional groups to environmental variables is extremely important in maintaining biodiversity in aquatic ecosystems. Although there were more taxonomic units at the species level than at the functional group level, it remained unclear whether species diversity was more sensitive than functional group diversity to environmental variables. In this study, taxonomic composition and alpha–beta diversity of phytoplankton were investigated in 23 subtropical reservoirs located in the Han River Basin in South China during wet and dry seasons. Structural Equation Modelling (SEM) and Generalized Dissimilarity Modelling (GDM) were employed to validate the response of phytoplankton species and functional group alpha–beta diversities to environmental variables. The results indicated that the community compositions of phytoplankton in eutrophic reservoirs were similar between wet and dry seasons, while there were distinct differences for community composition in oligotrophic–mesotrophic reservoirs between the two seasons. Across all reservoirs, there were no significant differences in alpha and beta diversities of species and functional groups between wet and dry seasons. The SEM and GDM results revealed that total phosphorus was the primary driving factor influencing alpha and beta diversities of species and functional groups in the 23 reservoirs. Meanwhile, the non-linear results of species beta diversity were stronger than the non-linear results of functional group beta diversity, indicating that phytoplankton species exhibited a higher explanatory power in responding to environmental changes compared to that of functional groups. Compared to that of species beta diversity, the response of functional group beta diversity to environmental variables was significantly lower in the dry season. These research findings lead to re-evaluating the common practice relating to the use of phytoplankton functional groups to assess environmental conditions, which may overlook the explanatory power of subtle changes at the species level, especially during periods of habitat diversification in the dry season.

## 1. Introduction

Exploring biodiversity, with a focus on alpha and beta diversity, is a vital aspect of ecological research. Alpha diversity refers to the diversity within a community of a specific area, representing the diversity characteristics within a certain spatial region. Beta diversity refers to the dissimilarity in species composition between communities in different habitats along an environmental gradient, or the rate of species turnover along an environmental gradient [1,2,3]. Beta diversity primarily arises from two processes: species replacement and the loss or addition of species. Species replacement between two habitats is also known as turnover, while non-random loss of species leads to a nested structure in species composition [4,5]. The relative contributions of turnover and nestedness to beta diversity reflect ecological processes underlying biodiversity maintenance. Differences in their contributions indicate varying dynamics in how species composition changes across habitats and in response to environmental and ecological factors [6,7]. Beta diversity reflects changes in community composition, and its response to environmental gradients can be analyzed using the “distance-based” approach. Previous studies mostly employ linear statistical methods like the Mantel test for statistical analysis [8]. However, due to interactions and indirect effects, species often exhibit non-linear responses to environmental changes. This complexity can lead to low explanatory power in Mantel regressions as these tests are based on linear assumptions and may not adequately capture the intricate and non-linear nature of ecological interactions and responses to environmental gradients [9]. Generalized Dissimilarity Modelling (GDM) utilizes generalized linear models and spline functions to address non-linear responses. This approach enables a more effective revelation of how beta diversity responds to environmental gradients and allows for the prediction of changes in community composition within a region [10,11].

As key primary producers in aquatic ecosystems, phytoplankton can rapidly respond to changes in environmental variables within freshwater ecosystems [12]. Earlier studies mainly concentrated on analyzing phytoplankton species composition, abundance, and diversity, however, the analysis of functional groups alongside taxonomic composition have been explored in the past 20 years [13,14]. This innovative classification approach categorizes phytoplankton species into a variety of functional groups based on species characteristics of habitat adaptability and sensitivity. The establishment of this functional group classification is based on two crucial ecological assumptions: Firstly, species within a certain group that have stronger adaptability are more capable of tolerating limiting factors in the environment than those with weaker adaptability. Secondly, a complex environment encompassing a variety of environmental factors is always occupied by a series of species with similar adaptabilities. Therefore, the status of water body can be elucidated by identifying the habitat types to which the dominant functional groups are adapted. This approach has previously been successfully applied to analyze phytoplankton communities in Southern China’s reservoirs, demonstrating its effectiveness in evaluating ecological status and environmental conditions of aquatic ecosystems with phytoplankton functional groups [15,16]. Although there are studies relating to beta diversity of phytoplankton species in response to environmental changes [17,18], little information is available about beta diversity of phytoplankton functional groups or the comparison between species and functional groups, despite being under the evidence line regarding the presence of different responses of phytoplankton species and functional groups to environmental factors [19]. It is, therefore, urgent to conduct a study focusing on the non-linear responses of both phytoplankton species and functional group beta diversity to environmental variables.

Compared with those in temperate regions, lakes and reservoirs located in the low-latitude south subtropical region, influenced by a monsoon climate, experience strong hydrological changes but little seasonal variation in water temperature. The relatively high temperature along a whole year may result in similar seasonal distribution in phytoplankton diversity [20,21]. It has previously been reported that nutrients are one of the key environmental factors influencing the composition of phytoplankton assemblages in the South Subtropical reservoirs [15]. In oligotrophic–mesotrophic reservoirs with relatively low nutrients, seasonal variations in rainfall significantly impact nutrient concentrations. During the wet season, increased rainfall leads to the input of higher external pollution and elevated nutrient concentrations in the reservoirs, while in the dry season, reduced rainfall leads to lower nutrient inputs. In eutrophic reservoirs, nutrient levels consistently remain high, regardless of wet and dry seasons. Eutrophic reservoirs maintain high nutrient levels throughout the year, with external nutrient inputs occurring during wet seasons and sediment release during dry seasons. Based on these observations, we hypothesize that phytoplankton community compositions in eutrophic reservoirs will be similar between dry and wet seasons, whereas in oligotrophic–mesotrophic reservoirs, there are significant differences in community composition between the two seasons. Based on the higher sensitivity of phytoplankton species to environmental changes compared to the functional group, and more taxonomic units at the species level than at the functional group level, we further hypothesize that the non-linear response of species beta diversity to environmental variables is expected to be greater than the response of beta diversity of functional groups.

## 2. Materials and Methods

### 2.1. Study Area

The Han River Basin (115°13′ to 117°09′ N and 23°17′ to 26°05′ E), located in the subtropical region of South China, has a length of 470 km and covers an area of 30,112 square kilometers. The Han River Basin is influenced by monsoon climate, characterized by hot and humid conditions with frequent heavy rain. The basin receives an average annual precipitation of 1600 mm, with the majority occurring from April to September. The survey was conducted in 23 reservoirs, including 4 large reservoirs and 19 medium-sized reservoirs, within the Han River Basin in August 2019 (wet season) and December 2019 (dry season). The locations of these 23 reservoirs are illustrated in Figure 1, and their morphological variables such as storage capacity and watershed characteristics are presented in Appendix A.

### 2.2. Sampling and Analysis

A water sample was collected in the area close to the dam from each of the 23 reservoirs. The environmental parameters, including water temperature (Temp), pH, electrical conductivity (Cond), and dissolved oxygen (DO), were measured in situ using a YSI-6600 portable analyzer. A liter sample of surface water was collected at 0.5 m depth and sent to the laboratory for analyzing total nitrogen (TN), nitrate nitrogen (NO_3_-N), nitrite nitrogen (NO_2_-N), ammonia nitrogen (NH_4_-N), total phosphorus (TP), and soluble reactive phosphorus (SRP) within 48 h after collection, following the procedures described in EPBC [22].

In the meantime, another one liter sample (0.5 m) of surface water was collected and fixed on-site using the Lugol’s iodine solution. After sedimentation for six hours, the concentrated sample was used for phytoplankton identification and abundance analysis. The phytoplankton taxa were identified to at least genus level, and to species level where possible. At least 500 algal cells were counted in each sample using a Sedgewick-Rafter chamber under an Olympus inverted microscope at 10 × 40 magnification [23].

### 2.3. Data Analysis

Four factors, including transparency, total nitrogen (TN), total phosphorus (TP), and chlorophyll-*a* (chl-a), were employed to evaluate the eutrophic state of water bodies. Chlorophyll-*a* was considered the baseline factor, and the other factors were weighted and integrated based on their degree of correlation with chlorophyll-*a* concentration. The trophic state of each reservoir was then calculated using this correlation-weighted composite index [24,25]. Nine of the twenty-three reservoirs were identified as eutrophic, while the remaining fourteen reservoirs were classified as oligotrophic–mesotrophic.

Following the functional group (FG) classification described in previous studies [13,14,26], phytoplankton were categorized into different functional groups. Seasonal variations of community and functional group compositions in trophic and oligotrophic–mesotrophic reservoirs were assessed through the analyses of Bray-Curtis dissimilarity matrices (ADONIS) and non-metric multidimensional scaling (NMDS) using the “vegan” package in R version 4.2.0.

For alpha diversity, the abundance and invsimpson diversity of phytoplankton species and functional groups in each of the reservoirs were calculated. Invsimpson diversity was computed using the “diversity” function in the “vegan” package in R. Beta diversity of species and functional groups between two seasons, alongside turnover and nestedness components, were calculated based on Bray-Curtis dissimilarities using the function “beta.pair.abund” in the “betapart” package in R [27].

Structural equation modelling (SEM) is an extension of the general linear model. Unlike multivariate regression analysis, SEM accounts for both the direct and indirect effects of latent variables on response variables. This approach also addresses potential measurement errors in structural relationships, a critical distinction from multiple regression analysis, which typically assumes the absence of measurement errors [28,29,30]. In this study, SEM played a crucial role in elucidating the direct and indirect factors influencing alpha diversity, particularly in terms of species and functional group abundance and the invsimpson index, as well as the dynamics of chlorophyll-*a* concentration. In the SEM analysis, reservoir capacity, water depth, and various physicochemical parameters, including water temperature, TN, and TP, were considered. SEM computations were performed using the ‘lavaan’ package in R.

To uncover the dominant factors influencing the beta diversity of species and functional groups, both linear (Mantel test) and non-linear (GDM) models were applied. The Mantel test was conducted using the “vegan” package in R, and significance was determined using 999 permutations. To address the two types of non-linearity in beta diversity, maximum likelihood estimation and monotonically increasing spline functions were used to transform the predictor variables. This approach allows for the modelling of the relationship between beta diversity among sampling sites and environmental gradients, as well as spatial distances. The GDM models, executed using the “gdm” package in R, incorporated three I-spline basis functions (mp = 3) for the non-linear fitting of environmental factors, thereby quantifying the relative impact of these variables on beta diversity variations [31].

## 3. Results

### 3.1. The Composition of Phytoplankton Species and Functional Groups

A total of 121 species of phytoplankton communities were identified in wet and dry seasons. Phytoplankton communities were mainly composed of green algae, accounting for 44% of the total, followed by cyanobacteria and diatoms, each contributing 11% during the wet season. Similar composition was found in the dry season. The dominant species included *Limnothrix redekei*, *Sphaerospermopsis* sp., *Cylindrospermopsis* sp., and *Microcystis* sp. in the wet season, while the phytoplankton community was dominated by *Limnothrix redekei*, *Pseudanabaena* sp., *Merismopedia* sp., and *Cylindrospermopsis* sp. in the dry season. In oligotrophic–mesotrophic reservoirs, phytoplankton abundance was obviously lower in the dry season than in the wet season, while in eutrophic reservoirs, there was no obvious difference in phytoplankton abundance between wet and dry seasons (Figure 2a,b).

Twenty-five functional groups were identified, including S1, M, MP, Lo, SN, J, X2, X1, F, P, X3, N, G, T, T_D_, T_B_, B, D, L_M_, Y, W1, W2, A, C, and E, with their corresponding environmental types detailed in Appendix A. The functional groups were dominated by 15 groups, as shown in Figure 2c,d. S1, Lo, MP, and F dominated the groups in the wet season, with S1 being the most abundant in seven reservoirs, while the groups S1, SN, J, P, and B were dominant in the dry season. Additionally, similar to the findings relating to species abundance, a lower abundance of functional groups was found in the dry season than in the wet season in oligotrophic–mesotrophic reservoirs.

The ADONIS and NMDS analyses revealed obvious differences between oligotrophic–mesotrophic and eutrophic reservoirs across both seasons, as evidenced by specific *p*-values for species and functional groups in oligotrophic–mesotrophic (*p*_species_ = 0.075, *p*_FG_ = 0.06) and eutrophic reservoirs (*p*_species_ = 0.625, *p*_FG_ = 0.897) (Figure 3). Similar phytoplankton community compositions between wet and dry seasons were found in eutrophic reservoirs, which was obviously different from the findings originating from oligotrophic–mesotrophic reservoirs.

### 3.2. Alpha and Beta Diversity

Because beta diversity refers to the dissimilarity in species composition among different habitats, the data of 23 reservoirs were combined for further analysis to reflect the changes in the environmental gradient. Compared to the wet season, the dry season exhibited a relatively higher species richness, yet the difference was not statistically significant. Species richness ranged from 8 to 32, with the peak value of 32 observed in the DFH reservoir during the dry season. The richness of functional groups exhibited a smaller variation between the two seasons, ranging from 5 to 14. Similarly, there were no significant differences for the invsimpson index calculated from both species and functional groups between the two seasons (Figure 4a). No significant difference was observed for beta diversity across the two seasons. Species turnover served as the dominant component in this context. Neither turnover nor nestedness components showed significant seasonal variations (Figure 4b).

### 3.3. Environmental Variables Influencing Alpha and Beta Diversity

In this study, four SEM models were developed to analyze phytoplankton species and functional groups with environmental variables in wet and dry seasons (Figure 5). In the wet season, total phosphorus (TP) and temperature (T) significantly affected species richness and chlorophyll-a levels, reflected by their significantly positive correlation relationships (Figure 5a). Furthermore, a significantly positive correlation (*R*^2^ = 0.336, *p* < 0.01) was observed between functional group richness and water temperature in the wet season (Figure 5c). In the dry season, species richness was significantly positively correlated with total phosphorus (*R*^2^ = 0.562, *p* < 0.001), but significantly negatively correlated with total nitrogen. Moreover, chlorophyll-a levels were significantly positively correlated with total phosphorus (*R*^2^ = 0.647, *p* < 0.001), but significantly negatively correlated with reservoir capacity and water temperature. A significantly negative correlation relationship was observed between total phosphorus and water depth (*R*^2^ = 0.322, *p* < 0.01; Figure 5b), indicating that TP mainly originated from sediment release. A significantly negative correlation occurred between functional group richness and water depth (*R*^2^ = 0.375, *p* < 0.01; Figure 5d). Additional information about these environmental variables can be found in the Appendix A.

The Mantel test revealed there were no significant correlations between species richness, functional group diversity, and environmental factors in dry season (Appendix A). However, the GDM results indicated that beta diversity could be non-linearly predicted through ecological distance, with better predictive accuracy in the wet season than in the dry season (Appendix A).

The GDM analysis quantified the total explained variance in beta diversity across different groups, reflected by 20% for the species in the wet season, 23% for functional groups in the wet season, 20% for the species in the dry season, and 2% for functional groups in the dry season. During the wet season, water temperature and conductivity served as the primary factors influencing species beta diversity. Water temperature, total phosphorus (TP), and total nitrogen (TN) played a significant role in shaping functional group beta diversity. In the dry season, total phosphorus and spatial distance were key drivers influencing species beta diversity. However, the explained variances for functional group beta diversity were relatively low, with total phosphorus as the predominant factor explaining 0.9% only (Table 1).

## 4. Discussion

### 4.1. Seasonal Dynamics of Phytoplankton Communities and Nutrient Levels

In the present study, we observed distinct seasonal variations in phytoplankton assemblage composition within South Asian subtropical reservoirs, which were closely associated with nutrient levels. Although it has been reported that temperature promotes the growth of cyanobacterial species from *Microcystis*, *Anabaenopsis*, and *Oscillatoria* [32], temperature did not play the dominant role in seasonal changes in phytoplankton diversity in these reservoirs, probably due to the relatively subtle temperature variations across the whole year in the subtropical region compared to in the temperate region. Compared to water temperature, the impact of rainfall on phytoplankton species and communities is undoubtedly important [33,34]. Although a large amount of rain may dilute the abundance of phytoplankton in the wet season [34], the present study demonstrated that the abundance of phytoplankton in the wet season was significantly higher than in the dry season, probably due to the inputs of nutrients for phytoplankton growth. Nutrients like phosphorus were critical in affecting cyanobacterial growth, which was also supported by previous studies [35,36,37]. In these low-latitude reservoirs, phytoplankton species composition and abundance are mainly affected by trophic levels and hydrodynamic factors [38], and this pattern is evident in the Han River Basin reservoirs.

In the studied eutrophic reservoirs, similar phytoplankton communities were found between wet and dry seasons. During the wet season, these reservoirs mainly receive nutrients from river runoff, while in the dry season, internal release from sediment is the main source of nutrients. The SEM results highlight the important role of total phosphorus in influencing phytoplankton alpha diversity and chlorophyll-*a* concentration. These findings are in agreement with those reported by Stanley et al. and Saito et al. [39,40]. The presence of aquaculture in some of the studied eutrophic reservoirs (e.g., DFH, MX, and SB) also contributes to nutrient enrichment through an input of nutrients from fish food and waste. The results are supported by a recent study relating to nutrient availability fostering phytoplankton communities [41]. Moreover, Purdie et al. illustrated how nutrient uptake in the subsurface chlorophyll maximum (SCM) layer can further stabilize phytoplankton communities [42]. In contrast, significant seasonal variations in phytoplankton composition were found in oligotrophic–mesotrophic reservoirs (Figure 2), which should be ascribed to the nutrient disparities between the wet and dry seasons, due to the influence of precipitation patterns and sediment release. Increased runoff during the wet season enhances nutrient input, especially in oligotrophic–mesotrophic reservoirs, leading to substantial changes in phytoplankton composition between the two seasons. This aligns with findings originating from similar aquatic ecosystems [43], demonstrating the profound impact of seasonal nutrient dynamics on phytoplankton communities.

### 4.2. Non-Linear Responses of Species and Functional Group Beta Diversity

Environmental selection and dispersal limitation are identified as key mechanisms influencing the patterns of beta diversity [44,45], while the contribution of environmental selection and dispersal limitation to phytoplankton beta diversity varies across different types of aquatic ecosystems [31,46,47]. Our study highlighted that environmental change serves as a primary factor influencing beta diversity across the reservoirs. SEM and GDM analyses indicated that environmental selection is a dominant factor influencing both species and functional group diversity in the subtropical reservoirs of South Asia. Notably, species beta diversity demonstrates a more pronounced non-linear response to environmental variables than functional group beta diversity, suggesting a higher sensitivity of individual species to environmental changes. Reynolds documented similar patterns in aquatic ecosystems, which is consistent with the present study [26]. Species within the phytoplankton community respond distinctly along environmental gradients, seeking habitats optimal for their survival. This behavior is influenced by varying sensitivities of species to environmental stressors, as noted by Lean et al. and O’Farrell et al. [48,49]. In particular, sensitive species under environmental stress are more susceptible to extinction, while tolerant species can adapt and survive under such stress [50].

Although some studies have suggested that classifying phytoplankton into functional groups (FGs) is less resource-intensive than species-level classification, making it more suitable for routine biomonitoring [34], the findings of this study indicate that species-level classification of phytoplankton provides a more comprehensive explanation of variations in beta-diversity compared to FGs. The greater species diversity than functional group diversity means that changes in species composition are more sensitive to subtle environmental changes. In contrast, functional groups, due to their broader classification, may not respond quickly to these subtle changes. This difference in response was particularly evident in the present study, where species beta diversity responded more sensitively to environmental variations than functional groups did.

The different responses can be attributed to the diverse ecological niches occupied by various phytoplankton species, leading to a more sensitive response in species-level beta diversity. Due to the numerous taxonomic units of species and the sensitivity of rare species to aquatic conditions, species serves as a good indicator reflecting minor environmental changes. In contrast, functional groups with broader classifications might not effectively capture these subtle environmental shifts. The classification of phytoplankton into functional groups, which groups together species with similar ecological niches, leads to a reduction in overall beta diversity. The increase of beta-diversity at the species level reflects not only the adaptation of phytoplankton species to reservoir ecosystems, but also species contributions from ecological drift across reservoirs [51], thus further enhancing their sensitivity to spatiotemporal environmental changes.

The response of beta diversity to environment variables also shows seasonal differences. Species beta diversity has a high explanatory rate for environmental variables in both wet and dry seasons, while functional group beta diversity is more sensitive to environmental changes in the wet season than in the dry season. During the dry season, dispersal limitation also plays a role in maintaining phytoplankton beta diversity. Compared to the wet season, the relative importance of dispersal limitation increases in the dry season, probably because dispersal through water flow is a major dispersal pathway for phytoplankton. In the wet season, with more rainfall and faster flow rates, the role of dispersal limitation is smaller [52,53].

### 4.3. Limitations and Future Studies

We analyzed the communities and functional groups of phytoplankton in 23 reservoirs in the Han River Basin. By analyzing the differences in phytoplankton communities and their responses to the environment during the dry and wet seasons, we obtained our hypothesis. Although the data collection period was short, we used two models demonstrating reliable and robust results relating to the factors affecting phytoplankton species and functional communities in 23 subtropical reservoirs located in South China. In future research, we will collect long-term data and investigate various environmental factors (e.g., rainfall, flow velocity, storage capacity, etc.) to conduct long-term analysis of the changes in phytoplankton and functional communities in the Han River Basin.

Although, the GDM and Mantel models usually exhibited relatively low explanatory power, which has also been reported in previous studies [31]. The low explanatory power present in such field work may be ascribed to the uncertainty of many environmental parameters affecting phytoplankton species and functional communities. We conducted research on the species and functional communities of phytoplankton in the Han River Basin, and have gained further understanding of the changes in the functional communities of phytoplankton in the Han River Basin. Future studies are warranted to study the variations in the phytoplankton community’s diversity and its relationship with more environmental parameters in climatic zones in China.

## 5. Conclusions

We discovered that the phytoplankton assemblage compositions in eutrophic reservoirs were consistent between the wet and dry seasons, whereas in oligotrophic–mesotrophic reservoirs, the compositions obviously varied during the two seasons. The species beta diversity exhibited a more sensitive non-linear response than functional groups, reflecting a greater sensitivity to subtle environmental changes. Further investigation is warranted to discern how much of this species-level diversity increase is limited by environmental factors relative to influence of stochastic processes. These results will improve our understanding of ecological mechanisms governing phytoplankton diversity, and offer valuable perspectives for biodiversity conservation and ecosystem management in subtropical aquatic environments. Such understanding is crucial for developing measures to maintain ecological balance, ensuring the sustainability of these vital aquatic environments. Given that certain phytoplankton species, such as some dinoflagellates and euglenas, may adopt a heterotrophic lifestyle under specific conditions, further studies are warranted to evaluate the potential influence of heterotrophy among phytoplankton groups and their distribution during wet and dry seasons.

## Figures and Tables

**Figure 1 microorganisms-12-01547-f001:**
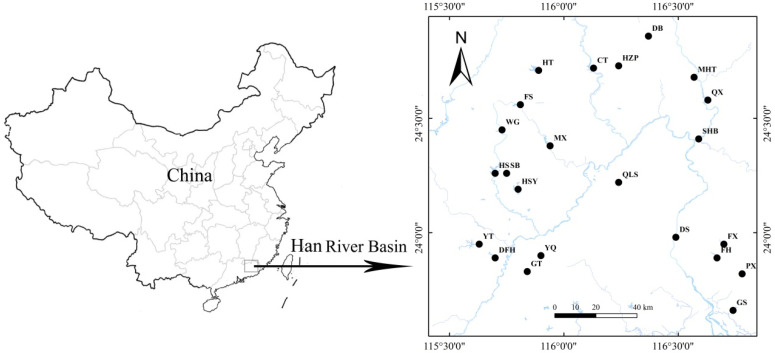
Location of 23 sampling reservoirs situated on the Han River Basin.

**Figure 2 microorganisms-12-01547-f002:**
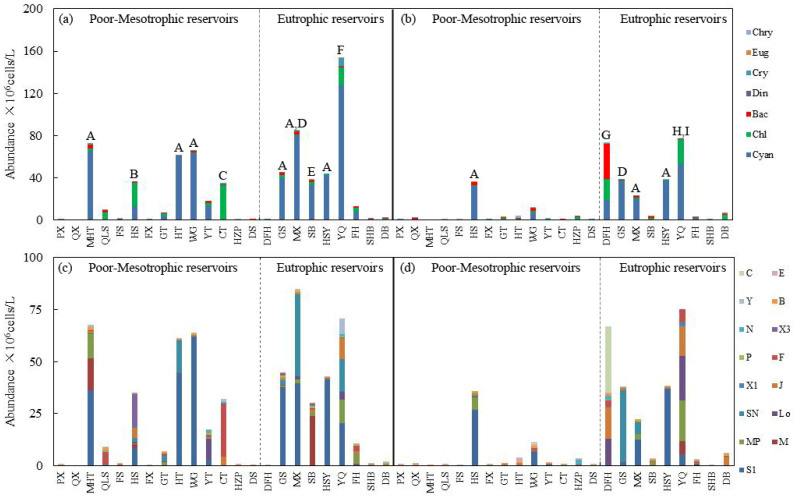
Abundance of phytoplankton species (**a**,**b**) and 15 dominant functional groups (**c**,**d**) of 23 reservoirs in wet (**a**,**c**) and dry (**b**,**d**) seasons, respectively. Dominant species: A. *Limnothrix redekei* B. *Chlorella vulgaris* C. *Sphaerocystis* sp. D. *Cylindrospermopsis* sp. E. *Microcystis* sp. F. *Sphaerospermopsis* sp. G. *Asterionella formosa* H. *Merismopedia* sp. I. *Pseudanabaena* sp. Abbreviation: Chry—Chrysophyta, Eug—Euglenophyta, Cry—Cryptophyta, Din—Dinophyta, Bac—Bacillariophyta, Chl—Chlorophyta, Cyan—Cyanobacteria.

**Figure 3 microorganisms-12-01547-f003:**
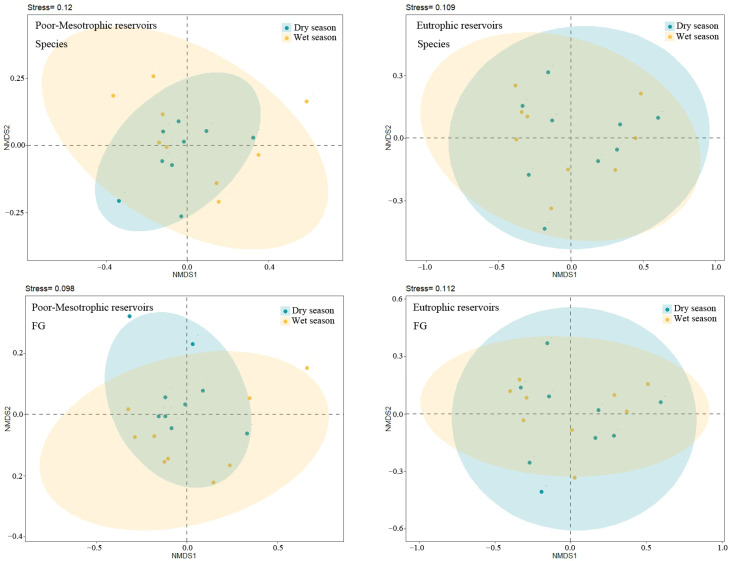
Non-metric multidimensional scaling (NMDS) ordination showing community composition patterns of species and functional groups (FG) in oligotrophic-mesotrophic and eutrophic reservoirs between wet and dry seasons.

**Figure 4 microorganisms-12-01547-f004:**
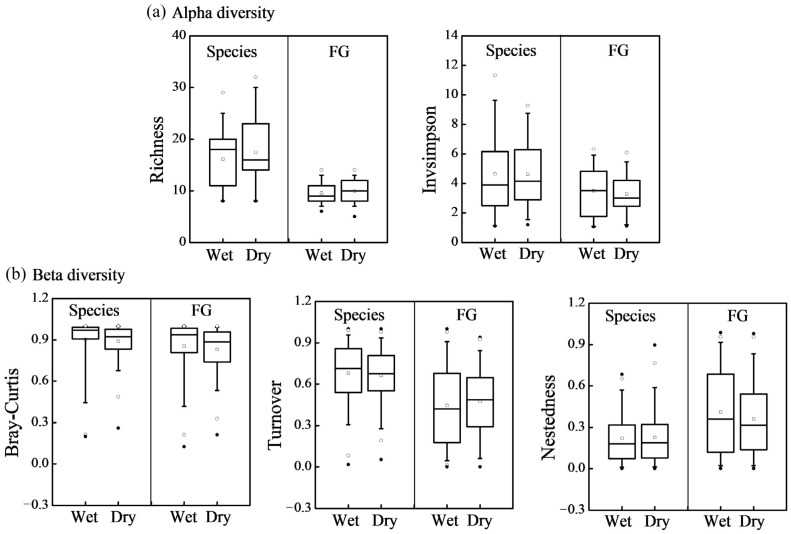
Alpha (richness and invsimpson) and beta diversity of phytoplankton species and functional groups in wet and dry seasons, respectively.

**Figure 5 microorganisms-12-01547-f005:**
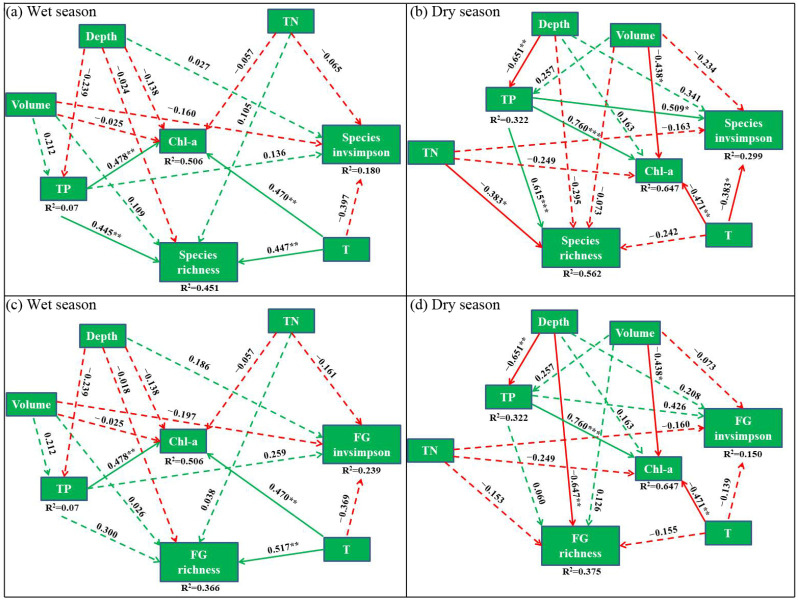
SEM models explaining phytoplankton diversity with environmental variables. The response of species richness and invsimpson index to environmental variables (**a**,**b**), the response of functional group richness and invsimpson index to environmental variables (**c**,**d**). FG: functional groups. TN: total nitrogen, TP: total phosphorus, Volume: reservoir capacity, T: water temperature, Chl-a: Chlorophyll-*a*. Arrows and numbers next to arrows indicate the effect size of the relationship and the associated boot strap *p*-value. * *p* < 0.05; ** *p* < 0.01; *** *p* < 0.001. The green and red lines represent positive and negative correlations, respectively.

**Table 1 microorganisms-12-01547-t001:** The fraction (%) of environmental factors explaining beta diversity in GDM models. TN: total nitrogen, TP: total phosphorus, T: water temperature, PO_4_: phosphate, NO_3_-N: nitrate, NH_4_-N: ammonia, Geo: geographic distance, Cond: conductivity.

Environmental Variables	Wet Season Percent Deviance Explained		Dry Season Percent Deviance Explained	
	Species	FG	Species	FG
**T**	7.87	5.97	/	/
**TN**	/	6.48	/	/
**TP**	0.75	4.91	6.73	0.9
**PO_4_**	/	1.65	/	/
**NO_3_-N**	3.25	2.54	2.22	/
**NH_4_-N**	2.7	1.48	2.62	0.54
**Geo**	0.57	/	5.14	0.36
**Cond**	4.88	0.47	4.06	0.47
**Total**	20.02	23.5	20.77	2.27

## Data Availability

All data will be made available on reasonable request.

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
