# Peer review of "Non-Linear Response of Alpha and Beta Diversity of Taxonomic and Functional Groups of Phytoplankton to Environmental Factors in Subtropical Reservoirs"

_microorganisms, 2024, doi:10.3390/microorganisms12081547_

Round 1

Reviewer 1 Report

Comments and Suggestions for Authors

Dear authors.

MS title could be more relevant to the content, perhaps “Non-linear answer of alpha and beta diversity based on taxonomic and functional groups of phytoplankton on environmental factors in subtropical reservoirs.”

Lines 13-17. Exploring the response of diversity to environmental variables was extremely important in maintaining biodiversity in aquatic ecosystems. This is actual problem, use Preset Simple (is).

Line 133 Need information about level of phytoplankton identification and which level data were used for calculations (in MS phytoplankton data presented at genus sp. Level only).

Line 189.  To illustrate differences you can have calculated similarity indexes between wet and dry seasons.

Lines 261-262. Remove these to the Discussion.

Line 299. “Purdie et al.” You have to correct the text font.

Author Response

Ms. Ref. No.: Microorganisms-3044571

A List of Point-by-point Responses to the Reviewers’ Comments

Reviewer #1:

1) MS title could be more relevant to the content, perhaps “Non-linear answer of alpha and beta diversity based on taxonomic and functional groups of phytoplankton on environmental factors in subtropical reservoirs.”

Authors’ response: Thanks for the suggestion. Revised as suggested.

2) Lines 13-17. Exploring the response of diversity to environmental variables was extremely important in maintaining biodiversity in aquatic ecosystems. This is actual problem, use Preset Simple (is).

Authors’ response: Revised as suggested in Line 14 in the revised version.

3) Line 133 Need information about level of phytoplankton identification and which level data were used for calculations (in MS phytoplankton data presented at genus sp. Level only).

Authors’ response: Thanks for the suggestion. The phytoplankton taxa were identified to at least genus level, and to species level where possible. We have added these information, shown as in Lines 136-137 in the revised version.

4) Line 189. To illustrate differences you can have calculated similarity indexes between wet and dry seasons.

Authors’ response: According to Figure 2, there is no obvious difference in the abundance of phytoplankton of the eutrophic reservoirs between wet and dry seasons. For the purpose of accurate and clear expressions, we have changed “significant” to “obvious” in Line 191 in the revised version.

5) Lines 261-262. Remove these to the Discussion.

Authors’ response: As suggested, we have moved this sentence to the Discussion section. Please refer to Line 321-323 in the revised version.

6) Line 299. “Purdie et al.” You have to correct the text font.

Authors’ response: Corrected as suggested in Line 307 in the revised version.

End of Our Responses

Reviewer 2 Report

Comments and Suggestions for Authors

The authors collected data during wet and dry seasons from 23 reservoirs in the Han River Basin of South China and analyzed the time series data using SEM and GDM. They explored the complex interactions between phytoplankton biodiversity and environmental variables and evaluated the impact of seasonal fluctuations on biodiversity through this comprehensive analysis.

However, the study has limitations including geographical constraints, limitations in functional group classification, and short data collection periods. Additionally, the models used present difficulties in interpreting the factors that affect phytoplankton communities and species succession.

While the attempt to interpret the dry and rainy seasons separately is commendable, the short-term nature of the survey makes accurate interpretation challenging. There is also a lack of discussion on the effects of rainfall.

Focusing primarily on a few environmental factors such as total phosphorus and water temperature presents a challenge that needs to be addressed. Future studies should involve long-term data collection, investigation of various environmental factors, and the development of models with higher explanatory power than complex nonlinear models such as GDM. Research at the phytoplankton species level is also necessary.

Given the floating nature of cyanobacteria, data collection by water layer is crucial. This aspect aligns with the authors' previous work, and significant efforts are required to enhance the quality of the paper.

The overall outline of the paper needs simplification. It would be more appropriate to treat this manuscript as a note rather than a full-length article.

Author Response

Ms. Ref. No.: Microorganisms-3044571

A List of Point-by-point Responses to the Reviewers’ Comments

Reviewer #2:

The authors collected data during wet and dry seasons from 23 reservoirs in the Han River Basin of South China and analyzed the time series data using SEM and GDM. They explored the complex interactions between phytoplankton biodiversity and environmental variables and evaluated the impact of seasonal fluctuations on biodiversity through this comprehensive analysis.

However, the study has limitations including geographical constraints, limitations in functional group classification, and short data collection periods. Additionally, the models used present difficulties in interpreting the factors that affect phytoplankton communities and species succession.

Authors’ response: We would like to thank the reviewer for his/her great effort in going through our manuscript and providing many constructive and useful comments.

We have carefully revised and improved our manuscript based on the suggestions and comments from the reviewer. In the meanwhile, we believe that our work is quite novel and important, with the use of two models demonstrating reliable and robust results relating to the factors affecting phytoplankton species and functional communities in 23 subtropical reservoirs located in South China.

While the attempt to interpret the dry and rainy seasons separately is commendable, the short-term nature of the survey makes accurate interpretation challenging. There is also a lack of discussion on the effects of rainfall.

Authors’ response: We appreciate this comment from the reviewer, and we have added the discussion on the effects originating from precipitation. Please refer to Lines 287-292, which is also shown below.

“Compared with water temperature, the impact of rainfall on phytoplankton species and communities is undoubtedly important [33-34]. Although a large amount of rain may dilute the abundance of phytoplankton in the wet season [34], the present study demonstrated that the abundance of phytoplankton in the wet season was significantly higher than that in the dry season, probably due to the inputs of nutrients for phytoplankton growth.”

Focusing primarily on a few environmental factors such as total phosphorus and water temperature presents a challenge that needs to be addressed. Future studies should involve long-term data collection, investigation of various environmental factors, and the development of models with higher explanatory power than complex nonlinear models such as GDM.

Authors’ response: We understand the concern from the reviewer. However, the GDM and Mantel models usually exhibited a little bit low explanatory power, which is also reported in previous studies (e.g., only 16% for the work by Xu et al., 2022). The low explanatory power present in such field work may be ascribed to the uncertainty of many environmental parameters affecting phytoplankton species and functional communities.

Cited references:

Xu, Y, Xiang Z, Rizo EZ, Naselli-Flores L, Han BP. Combination of linear and nonlinear multivariate approaches effectively uncover responses of phytoplankton communities to environmental changes at regional scale. Journal of Environmental Management, 2022, 305, 114399.

We have also added related information about future study directions at the end of the Conclusion section.

Research at the phytoplankton species level is also necessary.

Authors’ response: Thanks for the suggestion. Added as suggested in Lines 136-137.

Given the floating nature of cyanobacteria, data collection by water layer is crucial. This aspect aligns with the authors' previous work, and significant efforts are required to enhance the quality of the paper.

Authors’ response: We fully agree with the reviewer’s suggestion. We have added the information relating to sampling depth in Lines 128 and 133.

The overall outline of the paper needs simplification. It would be more appropriate to treat this manuscript as a note rather than a full-length article.

Authors’ response: We would like to thank the reviewer again for reviewing our manuscript, but we also believe our data are sufficient to support it to be published as a full-length research article. Also, our work is quite novel and important to describing the factor shaping phytoplankton communities in 23 reservoirs located in South China.

Reviewer 3 Report

Comments and Suggestions for Authors

This MS is evaluated as a ONE research study on the spatial heterogeneity of phytoplankton communities or species at 23 study sites during wet and dry season 2019. It seems to be somewhat insufficient material in terms of academic value. The data content is also very insufficient, and the research results that used the data in this manuscript to classify taxonomic and functional groups at the phytoplankton community or species level are assessed to have clear limitations.

 Overall, the content of the current manuscript is evaluated as insufficient to evaluate the results of experiments based on hypothesis testing or monitoring that includes a large amount of information as modeling results. Therefore, it is evaluated that this manuscript should undergo a reevaluation process based on a larger amount of information or data on phytoplankton from various lakes under different aquatic environmental conditions in climatic zones in China.

 In addition, the research results showing that some abiotic factors affect the distribution of phytoplankton are examples of various research results or paper data depending on the state of the water environment, and the originality of this manuscript is evaluated to be insufficient. Although it is not confirmed how many times the samping was conducted in 2019, the research results presented in this MS are considered to have limitations in quantitative evaluation. I inform you of the regrettable results, but that the MS cannot be reviewed in its current status.

Comments on the Quality of English Language

In general, the quality of English MS can be improved at the level of revision and editing of most MS, but in essence, it is believed that additional supplementary checks should be prioritized for the main quantitative evaluation of the results of this study.

Author Response

Ms. Ref. No.: Microorganisms-3044571

A List of Point-by-point Responses to the Reviewers’ Comments

Reviewer #3:

This MS is evaluated as a ONE research study on the spatial heterogeneity of phytoplankton communities or species at 23 study sites during wet and dry season 2019. It seems to be somewhat insufficient material in terms of academic value. The data content is also very insufficient, and the research results that used the data in this manuscript to classify taxonomic and functional groups at the phytoplankton community or species level are assessed to have clear limitations.

Overall, the content of the current manuscript is evaluated as insufficient to evaluate the results of experiments based on hypothesis testing or monitoring that includes a large amount of information as modeling results. Therefore, it is evaluated that this manuscript should undergo a reevaluation process based on a larger amount of information or data on phytoplankton from various lakes under different aquatic environmental conditions in climatic zones in China.

Authors’ response: We regret to receive the negative comments from the reviewer, but we would like to thank him/her for reading our manuscript. We think the reviewer may have overlooked our research objective and hypotheses (Lines 101-109 in the revised version). Our data collected from 23 reservoirs located in South China are sufficient to respond research objectives and to support our hypotheses.

Our work shows that phytoplankton community compositions in eutrophic reservoirs are similar between dry and wet seasons, whereas in oligotrophic-mesotrophic reservoirs, there are obvious differences in community composition between the two seasons. Also, the GDM model suggested that the non-linear response of species beta diversity to environmental variables is greater than the response of beta diversity of functional groups.

This work is of high quality, with sufficient data answering many important questions. The work is quite novel and important, with significant contributions to the research field of aquatic ecology.

We have also added related information about future study directions at the end of the Conclusion section.

In addition, the research results showing that some abiotic factors affect the distribution of phytoplankton are examples of various research results or paper data depending on the state of the water environment, and the originality of this manuscript is evaluated to be insufficient. Although it is not confirmed how many times the samping was conducted in 2019, the research results presented in this MS are considered to have limitations in quantitative evaluation. I inform you of the regrettable results, but that the MS cannot be reviewed in its current status.

Comments on the Quality of English Language

In general, the quality of English MS can be improved at the level of revision and editing of most MS, but in essence, it is believed that additional supplementary checks should be prioritized for the main quantitative evaluation of the results of this study.

Authors’ response: Again, we thank the reviewer for reading our work, but we believe our work is high novel and quite important, as mentioned above.

We have clearly provided sampling information like sampling date (August and December 2019), sampling sites and methods (Lines 112-121). Therefore, we don’t know what the reviewer means by saying “Although it is not confirmed how many times the samping was conducted in 2019…”

Round 2

Reviewer 2 Report

Comments and Suggestions for Authors

The authors responded relatively sincerely and proactively to the reviewers' advice. What is somewhat disappointing is that although their research has various limitations and limitations, efforts to generalize this to various readers are somewhat lacking. I think it is necessary to make some mention of the scale and limitations of one's research in the introduction or discussion. For this purpose, additional insertion of some related previous studies is necessary.

Author Response

Ms. Ref. No.: Microorganisms-3044571

A List of Point-by-point Responses to the Reviewers’ Comments

Reviewer #2:

The authors responded relatively sincerely and proactively to the reviewers' advice. What is somewhat disappointing is that although their research has various limitations and limitations, efforts to generalize this to various readers are somewhat lacking. I think it is necessary to make some mention of the scale and limitations of one's research in the introduction or discussion. For this purpose, additional insertion of some related previous studies is necessary.

Authors’ response: We appreciate this comment from the reviewer, and we have increased the discussion on the scale and limitations of our research. Please refer to Lines 366-385, which is also shown below.

We analyzed the communities and functional groups of phytoplankton in 23 reservoirs in the Han River Basin. And by analyzing the differences in phytoplankton communities and their responses to the environment during the dry and wet water periods, we obtained our hypothesis. Although the data collection period was short, we used two models demonstrating reliable and robust results relating to the factors affecting phytoplankton species and functional communities in 23 subtropical reservoirs located in South China. In future research, we will collect long-term data and investigate various environmental factors (e.g., rainfall, flow velocity, storage capacity,etc.) to conduct long-term analysis of the changes in phytoplankton and functional communities in the Han River Basin.

Although, the GDM and Mantel models usually exhibited a little bit low explanatory power, which is also reported in previous studies [31]. The low explanatory power present in such field work may be ascribed to the uncertainty of many environmental parameters affecting phytoplankton species and functional communities. We have conducted research on the species and functional communities of phytoplankton in the Han River Basin, and have gained further understanding of the changes in the functional communities of phytoplankton in the Han River Basin. Future studies are warranted to study the variations of phytoplankton community and diversity and their relationships with more environmental parameters in climatic zones in China.